# $\alpha$-Former: Local-Feature-Aware (L-FA) Transformer

**Zhi Xu**[1]      **Bin Sun**[1]      **Yue Bai**[1]      **Yun Raymond Fu**[1]

[1]Department of ECE, , Northeastern University, Boston, Massachusetts, USA

## Abstract

Despite the success of current segmentation models powered by the transformer, the camouflaged instance segmentation (CIS) task remains a challenge due to the target and the background are similar. To overcome this problem, we propose a novel architecture called the local-feature-aware transformer ($\alpha$-Former), inspired by how humans find the camouflaged instance in a given photograph. We use traditional computer vision descriptors to simulate how humans find the unnatural boundary in a given photograph. Then, the information extracted by traditional descriptors can be employed as prior knowledge to enhance the neural network's performance. Moreover, due to the non-learnable characteristics of traditional descriptors, we designed a learnable binary filter to simulate the traditional descriptors. In order to aggregate the information from the backbone and binary filter, we introduce an adapter to merge local features into the transformer framework. Additionally, we introduce an edge-aware feature fusion module to improve boundary results in the segmentation model. Using the proposed transformer-based encoder-decoder architecture, our $\alpha$-Former surpasses state-of-the-art performance on the COD10K and NC4K datasets.

## 1 INTRODUCTION

Camouflaged instance segmentation (CIS) is beneficial for applications in computer vision, like medical image segmentation, agriculture, etc ( Fan et al. [2020]). However, this task is challenging compared to traditional object detection and segmentation since camouflaged objects can effectively blend in with the background, making it difficult for models to detect and annotate them accurately.

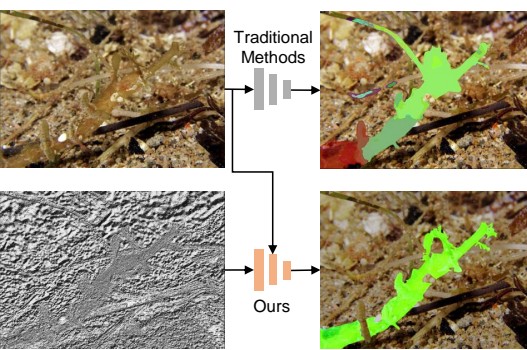

Figure 1: The $\alpha$-Former was motivated by the need to improve the performance of the camouflaged instance segmentation model. The model generates a local feature that provides precise boundary information about the target object. The input image is displayed in the top left, the prediction result without the local feature is shown in the top right, the generated local feature is displayed in the bottom left, and the prediction result with the local feature is shown in the bottom right. Incorporating the local feature into the model results in a more accurate segmentation of the target object.

Recently, transformer reached outstanding performances in different applications like detection ( Carion et al. [2020]), classification ( Chen et al. [2021]), segmentation ( Strudel et al. [2021]), etc. However, transformer models usually need a large-scale dataset for training. Thanks to the large-scale datasets and benchmarks for camouflaged object detection including , NC4K ( Lv et al. [2021]), COD10K ( Fan et al. [2020]), CAMO ( Le et al. [2019]), CAMO++ ( Le et al. [2021]), the researchers can implement the transformer on CIS. As a result, the transformer have achieved state-of-the-art performance in this field ( Pei et al. [2022]).

Despite their effectiveness, current transformer models have limitations in dealing with CIS. As shown in Fig 1, these models tend to predict multiple objects for a single target when the edge is unclear. This is because the models pri-

marily focus on finding the target object and ignore the importance of accurately identifying the boundary of the target object. To improve CIS performance, models need to understand the object's location better and enhance the features around the instance's boundary.

Inspired by how humans detect hidden objects within a photograph, the approach does not involve a direct search for the concealed instance due to the object's seamless integration into the surroundings, making it challenging to pinpoint directly. Instead, humans rely on comparing the local features with adjacent pixels. When humans recognize an unnatural boundary, it raises confidence in the presence of a concealed object (Troscianko et al. [2009]). However, the question arises of how to impart prior knowledge to a neural network regarding identifying unnatural boundaries. This is where traditional descriptors come into play. The fundamental concept behind these descriptors is to establish a means of comparing a pixel to its neighboring pixels. Illustrated in Fig.1, the lower left image demonstrates the outcome of applying a traditional descriptor to a camouflaged instance, revealing the ability of such descriptors to identify unnatural boundaries in the given image. Subsequently, this information can be employed as prior knowledge to enhance the neural network's performance.

To improve the boundary features of our model, we have integrated traditional descriptors like LBP ( Ojala et al. [1994]) into the transformer framework. LBP is especially sensitive to edges, which is advantageous in the context of CIS because of the high similarity between foreground and the background. As depicted in Figure 1, LBP can accurately demarcate the boundary of the target object, even when the texture and color of the target object have high similarity to that of the background. This makes it possible for the model to achieve superior results, as shown in Figure 1. By combining LBP with the transformer, we have developed an effective framework for identifying target objects and creating precise boundaries. We call this framework the local-feature-aware transformer, or $\alpha$-Former (pronounced "alpha-former"). Inspired by LBP, we have created a learnable module known as the binary filter (BF), which can compare pixel values within a field and generate a local feature. The binary filter consists of a learnable module and a fixed-weight convolution layer called BCNN which can extract features similar to the LBP.

The fixed convolution layer is able to generate local features by comparing different pairs of pixels, while the learnable module can collect and consolidate this comparison information. To effectively integrate the features extracted by the binary filter, we have developed a learnable module known as the feature aggregation adapter (FAA). The FAA can provide the local features to the backbone of the model without interfering with its performance, even if there are differences in the input distribution. Moreover, our FAA module is highly parameter-efficient and easy to train. Additionally,

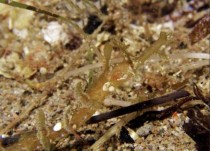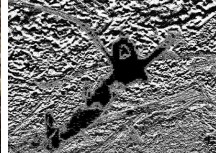

Figure 2: Examples of our BCNN layer. The left is a sample of $3 \times 3$ BCNN layer, the center is the input image, and the right is the output of the BCNN layer. Our BCNN is a fixed-weight binary convolution layer that can provide the comparison information of neighboring pixel pairs. Our results show that BCNN can provide a precise boundary for a given image.

we have designed an edge-aware module that can accurately predict boundaries for CIS. This module includes a multi-level convolution layer that offers a wide receptive field, as well as a fixed-weight convolution layer that extracts local features. To utilize the ground truth edge as supervision, we employ a $1 \times 1$ convolution layer to generate edge predictions. These edge predictions are then incorporated into the final prediction head to improve overall performance of the model.

Our model combines the binary filter (BF), feature aggregation adapter (FAA), and edge-aware fusion module to achieve superior performance on two popular datasets, NC4K and COD10K. Specifically, our architecture outperforms the current state-of-the-art by approximately 2 average precision (AP) points. Additionally, we do comprehensive ablation studies to demonstrate the effectiveness of the proposed BF, FAA, and edge-aware fusion module. Also, we provide lots of qualitative results in our experiments.

To summarize, our contributions are:

- Inspired by how human find the camouflaged instance in a photograph, we use traditional descriptors to simulate the process of how human find the unnatural boundary. Moreover, due to the non-learnable characteristics of traditional descriptor, we proposed a learnable module to extract similar features as the traditional descriptor.

- We proposed $\alpha$-Former, which firstly provides local binary information to the camouflage instance segmentation model. Also, we provide edge supervision to our model to improve the final mask boundary.

- We achieve state-of-art camouflaged instance segmentation results on two different benchmarks. Experiments and ablation study prove the efficiency of our proposed modules and architecture.

## 2 RELATE WORK

**Camouflaged Object Detection.** Camouflaged object detection aims to find the object in the image that hidden in the background and is more difficult than traditional object detection. Earlier works mainly focus on some level features like color (Huerta et al. [2007]), texture (Song and Geng [2010]). As deep learning advances, an increasing number of studies are employing neural networks to address the issue. These methods (including Zhu et al. [2021], Mei et al. [2021]) mostly employ a CNN backbone for high-level feature extraction and aim to detect and predict the camouflaged objects.. (Zhai et al. [2021]) proposed MGL that firstly use a mutual graph to detect and predict the final results. (Yang et al. [2021]) proposed UGTR which tried to mimic the human process, adding an uncertain prediction for camouflaged object detection. (Pei et al. [2022]) proposed OSFormer that uses a one-stage architecture and transformer to get the final results. (Mei et al. [2021]) introduced PFNet that firstly adds a focus and positioning module to mimic the human detection process, which tries to find the target object.

**Integrating traditional descriptors to Help CNN.** There is a long history of using traditional descriptors to help improve the performance of CNN. Earlier works use different descriptors to help CNN. For example, some works (Karanwal and Diwakar [2021a,b]) use LBP (Ojala et al. [1994]) to help improve the face performance recognition. People also use HOG (Dalal and Triggs [2005]) to help them improve the performance of human detection (Surasak et al. [2018]) and action recognition (Patel et al. [2020]). Recently, researchers tried to combine SIFT (Lowe [1999]) and convolution networks (including Gupta et al. [2019], Hossein-Nejad et al. [2021], Kovač and Marák [2022]) to extract better features and implement the features in different applications. Considering so many works integrating traditional descriptor with deep learning architecture and get performance improvement and the lack of effort to apply the traditional descriptor to camouflaged object detection, we try to use a descriptor inspired by traditional descriptors to enhance the effectiveness of camouflaged object detection.

**Binary Filter.** The traditional descriptor inspires the idea of using a binary filter for convolution. Many works already use their binary filter to get good performance in many datasets. For example, BinaryConnect (Courbariaux et al. [2015]) tried to design a neural network that only has binary weights in propagation. In this article, they approximate the real value in neural networks with binary values. Based on BinaryConnect, (Courbariaux et al. [2016]) proposed BinaryNet where both the activations and weights are constrained to $-1$ or $+1$. LBCNN (Juefei-Xu et al. [2017]) uses a fixed-weight binary convolution to replace the original convolution and get good performance in the classification tasks. These works show the feasibility of using binary fil-

ters to extract features and train neural networks.

**Adapter.** The adapter is firstly proposed in NLP tasks (Houlsby et al. [2019]), which targets to transfer the pre-trained NLP model to different downstream tasks while not introducing lots of parameters in the new models. Because of its efficiency, more and more researchers have recently tried to add an adapter to computer vision tasks (including Long Li et al. [2019], Sung et al. [2022]). Also, it is very efficient to use adapter in domain transfer, and lots of works (including Ansell et al. [2021], Ke et al. [2021]) that concentrate on this. The input domain has changed after using the binary filter in our work. Hence, we use an adapter to help the pre-trained backbone to extract the features.

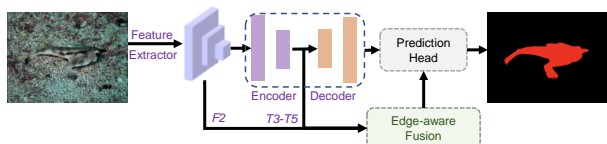

Figure 3: $\alpha$-Former comprises a feature extractor, an encoder-decoder, an edge-aware fusion module, and a prediction head. $\alpha$-Former use a single RGB image as input, and $\alpha$-Former output the camouflaged object mask in the input image.

## 3 BINARY FILTER

### 3.1 WHY USE BINARY FILTER

We have observed that traditional camouflage segmentation models struggle to accurately determine the boundary of objects in ambiguous cases. For example, when presented with an image of a pipefish, as shown in Fig. 1, a standard model may predict multiple objects instead of correctly identifying the single target object. Inspired by how human find camouflaged instance, traditional descriptors come into our minds. We can use traditional descriptors to provide prior knowledge to neural network to enhance its ability. However, the traditional descriptors like LBP is not learnable, meaning that it is difficult to adapt to new input data.

To address this issue, we sought to design an architecture that can detect local binary features similar to those captured by traditional descriptors but is also learnable. The LBP descriptor compares the center pixel value with the surrounding pixel values, so we were inspired to create a binary filter using a fixed binary weight convolution (BCNN) to simulate this process.

### 3.2 ARCHITECTURE OF BINARY FILTER

We describe the architecture of the proposed binary filter, which allows for comparison operations that are difficult to perform with traditional convolution layers in this section.

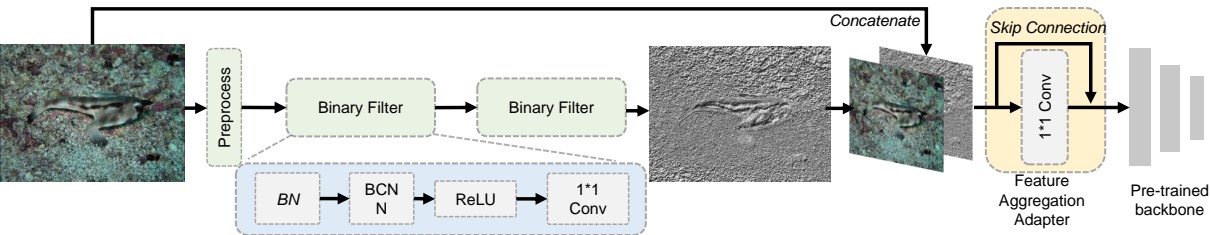

Figure 4: Our feature extractor contains a preprocessing module, several binary filters, a feature aggregation adapter, and a pre-trained backbone. The binary filter can extract local features of the input image. After getting local features, we concatenate the original image and local features and use our feature aggregation adapter to transfer the new image domain to the input image domain. After the feature aggregation adapter, we use a pre-trained CNN to do high-level and low-level features extraction.

As illustrated in Fig. 2, we can simulate the comparison operation by designing a convolution kernel where the center value is -1, the left value is 1, and all other values are 0. After applying this convolution operation, we compare the output with 0. If the output is greater than 0, we know that the value of left pixel is greater than the value of the center pixel; otherwise, the value of the left pixel is less than the value of the center pixel. Our designed binary convolution layer with fixed binary weight convolution (BCNN) can extract the precise boundary for the target object, as demonstrated in Fig. 2. To increase the robustness of BCNN, we use multiple binary convolution kernels for each BCNN layer, and for each kernel, we randomly select a value from $-1, 0, 1$. However, the BCNN is not trainable, and to make the binary filter trainable, we add a $1 \times 1$ convolution layer after each BCNN layer to gather information, which is trainable. This trainable $1 \times 1$ convolution layer is very light and easy to train compared to the traditional CNN architecture.

## 4 METHODS

**Architecture** Our proposed $\alpha$-Former has five crucial modules. (1) A feature extractor with a binary filter to extract similar features as the LBP ( Ojala et al. [1994]), an adapter to transfer the input domain, and a backbone to extract object features. (2) A transformer encoder that uses global and local features to generate object embedding. (3) An edge-aware feature fusion module to generate precise boundaries. (4) A transformer decoder to extract the information from the embedding (5) Mask predict head to predict final instance mask. The whole architecture is shown in Fig.3

### 4.1 FEATURE EXTRACTOR

Our feature extractor consists of three parts: a learnable local binary filter (BF), a feature aggregation adapter (FAA), and a pre-trained CNN backbone. These components are shown in Fig.4.

#### 4.1.1 Binary Filter (BF)

The purpose of the binary filter is to get local features. Here, provided an input image $I \in \mathbb{R}^{H \times W \times 3}$, we firstly use a convolution layer to preprocess the image. After the preprocessing, we use a pre-defined binary filter to extract local binary features. The detail of the binary filter is already discussed in Sec.3. We use multiple binary filters in every experiment to extract the local binary information. After the BF module, we can get a feature $F \in \mathbb{R}^{H \times W \times C}$ where channel number of the final $1 \times 1$ convolution is C. Then we concatenate the feature $F$ and the original image $I$.

#### 4.1.2 Feature Aggregation Adapter (FAA)

After the BF module, the channel numbers of concatenate images are different from the backbone training images, which makes it not practical to use the pre-trained backbone directly. To use the pre-trained backbone, we need a method to transfer the concatenated image domain to the domain that is the same as the original images. Here, we introduce a feature aggregation adapter to align the new image domain with the original image domain. The architecture of the adapter is a $1 \times 1$ convolution and a skip connection which can be seen in Fig.4. The adapter output a image with the shape $H \times W \times 3$ which is the same as the original images. The purpose of adding a skip connection is that, at the beginning of the training, it is challenging to initialize the parameter of the $1 \times 1$ convolution to guarantee the domain of the output is the same as the domain of the original image. In order not to influence the performance of the backbone at the beginning of the training, we can set very tiny initial values of the $1 \times 1$ convolution layer. Furthermore, for the skip connection, we can directly add the first three channels, the original images, to the output. This operation can ensure the input of the backbone is almost the same as the original image at the beginning of the training. Throughout the training phase, the model can gradually learn to use the local binary features.

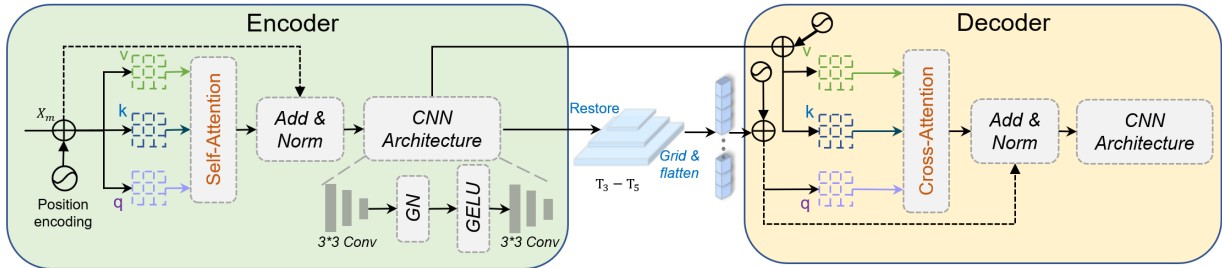

Figure 5: Our encoder contains a position encoding module, a self-attention module, and a CNN architecture. The encoder's input is the extracted third to fifth layer's backbone features. After getting the input feature, we integrate a position embedding to the features and use a self-attention module to get its local features. After getting the local feature, we use an add & norm operation followed by a CNN architecture to get the final output of the encoder. Then we restore and grid the output of the encoder to a location-aware query and input the query to the decoder. In the decoder, we use a cross-attention module to extract information. We use the same CNN architecture as the encoder after the cross-attention.

### 4.1.3 CNN BACKBONE

We use a pre-trained backbone in our experiments. In order to provide high-level features and low-level features to the prediction module, We utilize multi-scale features derived from the backbone. We will use the last four layers' features in most of our experiments. We will use $F_2 - F_5$ to represent different layer features in the following part. Because the backbone's input contains more local features than the original image, the extracted features of the backbone contain extra information compared to directly inputting the original images to the backbone.

### 4.2 ENCODER-DECODER

To speed up the training process and reduce the computation cost we combine the transformer and CNN in our encoder, which can be seen in Fig.5. We input multi-scale features $F_3 - F_5$ to our encoder to generate more informative features. Inspired by DETR (Carion et al. [2020]), which adds a position embedding to the input feature, We firstly calculate the position embedding of the input features and incorporate it into the original features $F3 - F5$ and get updated features $F3^{(1)} - F5^{(1)}$. Then we input the features to a self-attention module, which can capture the local information and get $F3^{(2)} - F5^{(2)}$. After the self-attention module, we use a CNN module to increase the training process. We add the features $F3^{(1)} - F5^{(1)}$ and $F3^{(2)} - F5^{(2)}$, then we pass the result of the self-attention module to a layer normalization, then we pass the result to a $3 \times 3$ convolution layer. After the convolution, A group normalization and a GELU activation are used. Following the GELU activation, we add a $3 \times 3$ convolution layer. After the convolution layer, we restore the outputs to multi-scale features $T_3 - T_5$. Then we flatten the $T_3 - T_5$ to a sequence and input them to the decoder. The decoder is the same as the encoder. We also combine the transformer and the convolution. For the input sequence,

we follow the same operation as the encoder, which first calculates the location embedding of the input features. After that, we grid the input sequence to the shape of $S \times S \times D$, then flatten them to query shapes $L \times D$ and produce a location-aware query that will provide the location information for every token. After getting the location-aware query, we input the encoder feature and location-aware query to a cross-attention layer. We use the encoder feature as the key and value, and use the location-aware query as the query in the cross attention layer. After the cross-attention layer, we use the same normalization layer and convolution structure as the encoder to produce the decoder embedding.

### 4.3 EDGE-AWARE FEATURE FUSION MODULE (EAF)

To improve the performance of boundary prediction, we added a module called edge-aware feature fusion. This module uses the ground truth edge as a guide to combine two types of features: high-level features extracted from the backbone network (called $F_2$) and low-level features extracted from the encoder (called $T_3$ to $T_5$).

The edge-aware feature fusion module processes the low-level features $T_5$ to $T_3$ by first extracting information with a convolution layer, then a binary convolutional layer is utilized to capture local binary features, which are then fed into a $1 \times 1$ convolutional layer to predict edges (called $E_5$).

Next, We up-sample the binary features to ensure they are the size is the same as $T_4$ and concatenate them, generating a new input feature ($I_4$). We repeat this process until we reach $F_2$.

Employing the edge-aware feature fusion module enables the model better recognize the boundaries of objects, leading to more precise segmentation masks and avoiding the issue of predicting one object as multiple objects. The formula for the edge-aware fusion model is given, and the output of

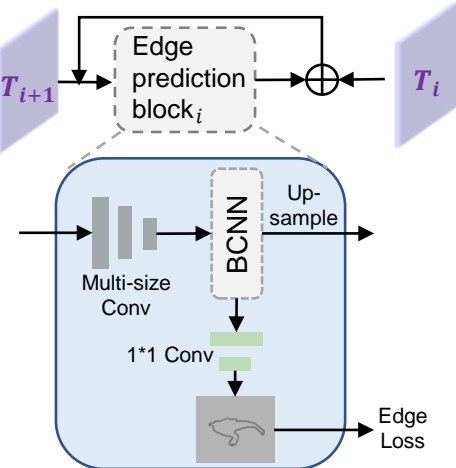

Figure 6: Our edge-aware feature fusion module uses a pyramid structure. The main component of our edge-aware feature fusion module is an edge prediction block. Given the input feature, we use a multi-size convolution following a BCNN layer to extract its feature. Then we up-sample the result to ensure that the size is the same as the next input feature size. We employ a $1 \times 1$ convolution layer to predict the edge and use the ground truth edge as supervision.

the final block $O_2$ is forwarded to the mask prediction head.

We also output the result of the final block $O_2$ to the mask prediction head.

## 4.4 MASK PREDICTION HEAD

We follow the same structure as OSFormer (Pei et al. [2022]). For more details, please see supplementary materials.

## 4.5 LOSS FUNCTION

Our loss function is composed of three parts, edge loss, location loss, and mask loss. We use dice loss for the edge loss and location loss; for mask loss, we use focal loss. Hence, our final loss function can be written as

$$L = \lambda_{edge}L_{edge} + \lambda_{location}L_{location} + \lambda_{mask}L_{mask}$$

. In our experiments, $\lambda_{edge}$ and $\lambda_{location}$ is set to 1 while $\lambda_{mask}$ is set to 3 to balance different loss.

## 5 EXPERIMENTS

### 5.1 EXPERIMENTAL SETUP

**datasets** We use two benchmark datasets: NC4K (Lv et al. [2021]) and COD10K (Fan et al. [2020]) in our experiments.

Table 1: Quantitative results of the $\alpha$-Former, the best results are highlighted in **bold**.

| method | COD10K | | | NC4K | | |
|---|---|---|---|---|---|---|
| | AP | AP50 | AP75 | AP | AP50 | AP75 |
| Mask-RCNN (He et al. [2017]) | 25.0 | 55.5 | 20.4 | 27.7 | 58.6 | 22.7 |
| MS-RCNN (Huang et al. [2019]) | 30.1 | 57.5 | 25.7 | 36.1 | 68.9 | 33.5 |
| Cascade RCNN (Cai and Vasconcelos [2019]) | 25.3 | 56.1 | 21.3 | 29.5 | 60.8 | 24.8 |
| HTC (Chen et al. [2019]) | 28.1 | 56.3 | 25.1 | 29.8 | 59.0 | 26.6 |
| Mask Transfiner (Ke et al. [2022]) | 28.7 | 56.3 | 26.4 | 29.4 | 56.7 | 27.2 |
| YOLACT (Bolya et al. [2019]) | 24.3 | 53.3 | 19.7 | 32.1 | 65.3 | 27.9 |
| CondInst (Tian et al. [2020]) | 30.6 | 63.6 | 26.1 | 33.4 | 67.4 | 29.4 |
| QueryInst (Fang et al. [2021]) | 28.5 | 60.1 | 23.1 | 33.0 | 66.7 | 29.4 |
| SOTR (Guo et al. [2021]) | 27.9 | 58.7 | 24.1 | 29.3 | 61.0 | 25.6 |
| SOLOv2 (Wang et al. [2020]) | 32.5 | 63.2 | 29.9 | 34.4 | 65.9 | 31.9 |
| OSFormer (Pei et al. [2022]) | 41.0 | 71.1 | 40.8 | 42.5 | 72.5 | 42.3 |
| $\alpha$-Former(Ours) | **42.5** | **72.8** | **41.8** | **42.9** | **72.9** | **43.3** |

The COD10K datasets include 3040 training images with instance-level annotations and 2026 for testing. The NC4K datasets contain 4121 images with instance-level labels. We train our model using the COD10K training set and test our model on COD10K testing set and NC4K dataset. In order to provide more training samples for the model, we resize the input images to multiple sizes. We ensure that the shorter side measures between 480 and 800 pixels, while the longer side of the input image is under 1333 pixels after resizing.
**evaluation metrics** We use COCO-style evaluation metrics in our experiments, including $AP$, $AP_{50}$ and $AP_{75}$, but our experiments have slight differences. The original COCO evaluation metrics use mAP, which will calculate the mean AP for every category. However, our camouflaged datasets are class-agnostic. Hence, we only need to calculate the AP for the whole dataset while ignoring the category.
**implement details** Pytorch is used to implement our $\alpha$-Former and we trained it on a single V100-sxm2. To build our model, ResNet-50 (He et al. [2016]) is used as the backbone, which had been trained with the ImageNet (Deng et al. [2009]) dataset. During our experiments, we trained our model for 90K iterations, utilizing a batch size of 2. The optimizer we used was SGD, the initial learning rate is $2.5e-4$, and the learning rate was reduced by a factor of 0.1 when the number of iterations reached 60K and 80K. The weight decay parameter is $1e-4$.

## 5.2 COMPARISON WITH THE STATE-OF-THE-ARTS

We conduct experiment to compare our model with current State-of-the-arts models. Because there are not many camouflaged instance segmentation models, we also use several generic instance segmentation models and limit these models to train and test on the camouflaged datasets. To have fair comparisons, pre-trained ResNet-50 was used as the backbone for all models. The results are shown in Table.1

Table 2: Comparison with the traditional descriptor, the best results are highlighted in **bold**.

| method | COD10K | | | NC4K | | |
|---|---|---|---|---|---|---|
| | AP | AP50 | AP75 | AP | AP50 | AP75 |
| Baseline | 40.244 | 69.875 | 39.422 | 41.718 | 71.640 | 41.179 |
| HOG | 40.934 | 70.887 | 40.285 | 42.765 | 71.988 | **44.226** |
| LBP | 40.410 | 70.323 | 40.184 | 41.794 | 71.313 | 42.484 |
| Circle-LBP | 40.424 | 69.622 | 40.764 | 41.921 | 71.661 | 42.133 |
| Binary filter | **42.453** | **72.735** | **41.758** | **42.936** | **72.905** | 43.278 |

## 5.3 ABLATION STUDY

### 5.3.1 Comparison with the traditional descriptor

As shown in Table.2, the performance of our binary filter and the traditional descriptor is compared. Here, Baseline means no descriptors are added. Because SIFT cannot generate a feature map with the same size as the input images, in order to use the same architecture and have a fair comparison, we mainly focus on the LBP (Ojala et al. [1994]), HOG (Dalal and Triggs [2005]), circle-LBP (Ojala et al. [2002]) descriptor in our experiments. Except for the local feature extractor, our experiments' other settings are the same. We can see that some of the traditional descriptors can outperform the model that does not include any local feature extractor. However, our learnable binary filter can perform better than the traditional descriptor. This experiment demonstrates our binary filter's efficiency and ability to provide powerful local features to enhance the model's performance.

### 5.3.2 Adapter

In this section, we show the improvement of adding the feature aggregation adapter to our feature extractor. The target for our adapter is to provide the extra local feature to our encoder. If we directly delete the adapter, the input domain will be different, and the pre-trained backbone cannot deal with the input with the local feature. However, to provide a fair comparison, we still need to provide the local feature to the encoder-decoder and the edge-aware fusion module. Hence, we concatenate our local features to the ResNet extracted features and change the input channel numbers of the encoder and edge-aware fusion module. In this way, we can still provide the local features to the encoder and edge-aware fusion module and provide a fair comparison. Also, we try a different setting that modified the first layer of the pre-trained backbone and randomly initialized (RI) this layer to demonstrate the efficiency of our adapter. To better show the effectiveness of our adapter, We also test the adapter on the traditional descriptor. The results are shown in Table.3. Noticed that our adapter is helpful for the binary filter and can improve the performance of the traditional descriptor.

Table 3: Ablations for the existence of feature aggregation adapter.

| method | COD10K | | | NC4K | | |
|---|---|---|---|---|---|---|
| | AP | AP50 | AP75 | AP | AP50 | AP75 |
| HOG + RI | 36.785 | 63.585 | 37.906 | 35.474 | 64.150 | 37.246 |
| HOG w/o adapter | 40.801 | 70.435 | 41.407 | 42.682 | 72.647 | 43.154 |
| HOG w/ adapter | 40.934 | 70.887 | 40.285 | 42.765 | 71.988 | 44.226 |
| LBP + RI | 33.562 | 61.623 | 34.732 | 35.631 | 64.463 | 35.462 |
| LBP w/o adapter | 39.530 | 69.419 | 39.331 | 42.288 | 71.077 | 42.162 |
| LBP w/ adapter | 40.410 | 70.323 | 40.184 | 41.794 | 71.313 | 42.484 |
| Circle-LBP + RI | 35.246 | 66.352 | 36.462 | 36.853 | 67.432 | 35.241 |
| Circle-LBP w/o adapter | 40.270 | 70.550 | 40.257 | 42.668 | 73.669 | 42.172 |
| Circle-LBP w/ adapter | 40.424 | 69.622 | 40.764 | 41.921 | 71.661 | 42.133 |
| Binary filter + RI | 36.415 | 64.151 | 35.414 | 33.541 | 67.252 | 34.532 |
| Binary filter w/o adapter | 41.427 | 71.247 | 40.984 | 42.610 | 71.517 | 42.985 |
| Binary filter w/ adapter | 42.453 | 72.735 | 41.758 | 42.936 | 72.905 | 43.278 |

Table 4: Ablations for the existence of edge-aware feature fusion module.

| method | COD10K | | | NC4K | | |
|---|---|---|---|---|---|---|
| | AP | AP50 | AP75 | AP | AP50 | AP75 |
| HOG w/o EAF | 37.658 | 66.584 | 35.984 | 39.252 | 67.971 | 38.756 |
| HOG w/ EAF | 40.934 | 70.887 | 40.285 | 42.765 | 71.988 | 44.226 |
| LBP w/o EAF | 36.128 | 67.197 | 36.725 | 36.375 | 68.258 | 37.813 |
| LBP w/ EAF | 40.410 | 70.323 | 40.184 | 41.794 | 71.313 | 42.484 |
| Circle-LBP w/o EAF | 35.254 | 64.741 | 36.194 | 36.581 | 66.943 | 36.135 |
| Circle-LBP w/ EAF | 40.424 | 69.622 | 40.764 | 41.921 | 71.661 | 42.133 |
| Binary filter w/o EAF | 38.019 | 69.765 | 36.813 | 37.083 | 68.672 | 38.731 |
| Binary filter w/ EAF | 42.453 | 72.735 | 41.758 | 42.936 | 72.905 | 43.278 |

### 5.3.3 Edge-aware feature fusion module

We provide the ablation study of our edge-aware fusion module in this section. Our edge-aware fusion module can provide precise boundary prediction information to the final prediction heads. We show the results using different descriptors, including traditional descriptors and our binary filter which is similar to the adapter. The results are shown in Table.4. Noticed that our proposed edge-aware feature fusion module can improve the performance for about 4 AP higher than the model do not have an edge-aware feature fusion module. It shows the efficiency of our edge-aware feature fusion module and proves that edge prediction is crucial in camouflaged instance segmentation. The qualitative results of our edge-aware feature fusion module can be seen in Fig.7, which shows that our edge-aware feature fusion module can deal with different situations and precisely predict the edge of the target object.

### 5.3.4 influence of different kernel size in BCNN

we investigate the impact of various kernel sizes on our binary filter in this section. Different kernel sizes will have different receptive fields, and a larger receptive field will provide more pixels in one convolution operation. In our binary filter, it will affect the final local binary feature of the binary filter. Our results are shown in Table.5. It shows

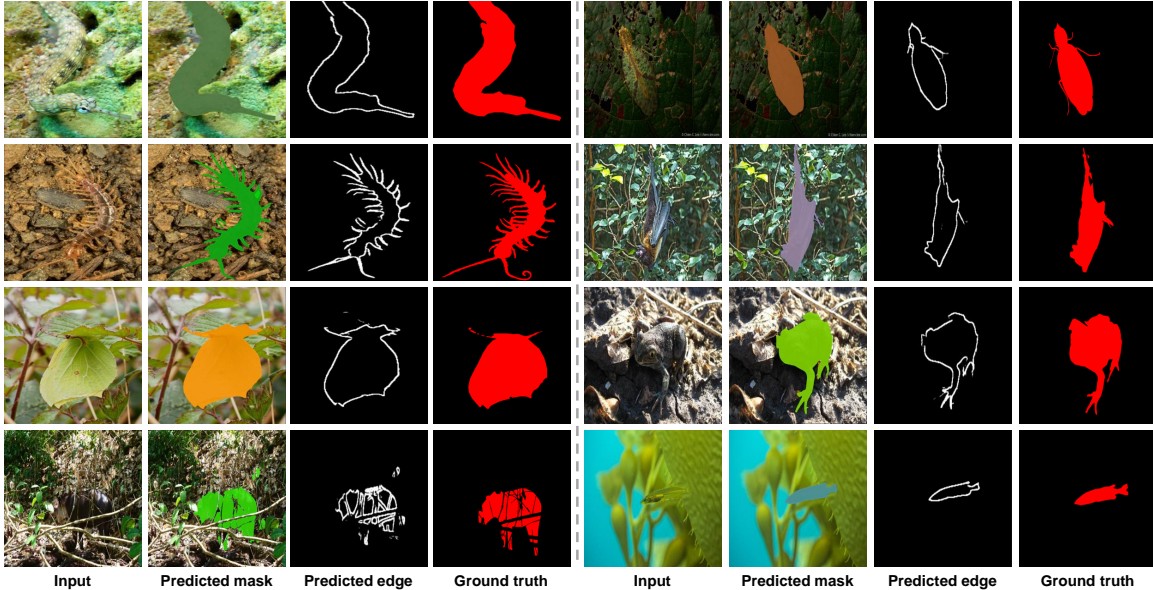

| Input | Predicted mask | Predicted edge | Ground truth | Input | Predicted mask | Predicted edge | Ground truth |

Figure 7: The results of our $\alpha$-Former's qualitative evaluation demonstrate its ability to extract precise boundaries and its strong performance in a range of challenging scenarios. These findings suggest that our proposed approach can effectively address the complexities of real-world image segmentation tasks.

Table 5: performance of $\alpha$-Former with different kernel size in the binary filter, the best results are highlighted in **bold**.

| method | COD10K | | | NC4K | | |
|--------|--------|------|------|------|------|------|
| | AP | AP50 | AP75 | AP | AP50 | AP75 |
| $3 \times 3$ | **42.453** | **72.735** | **41.758** | **42.936** | **72.905** | **43.278** |
| $5 \times 5$ | 41.308 | 70.624 | 41.707 | 42.567 | 72.075 | 43.198 |
| $7 \times 7$ | 40.476 | 70.047 | 40.790 | 42.136 | 71.895 | 42.698 |
| $9 \times 9$ | 40.691 | 70.116 | 40.810 | 41.164 | 71.043 | 42.580 |

that a smaller kernel size can have better performance. The reason that small kernel sizes have better performance may be that camouflaged objects have similar pixel values as the background. The larger kernel may increase the influence of the background and result in final performance drops.

### 5.4 VISUALIZATIONS

We presents the qualitative results of the $\alpha$-Former in this section, including the edge prediction achieved by our edge-aware fusion module. The results show the efficiency of our method, as our module can predict precise boundaries, as shown in the second row's first column, where it accurately identifies the feet of a challenging target object. Additionally, our $\alpha$-Former can successfully handle different backgrounds, such as branches, land, or aquatic plants, and precisely segment different target objects, including birds, fishes, and terrestrial animals. Moreover, our model can generate accurate edges even when the target object is partially occluded by the background, as seen in the last row's first column. This suggests that our approach can ex-

tract semantic information from the backbone's features and still recognize the object as the same entity, even if it is not continuous. Overall, these results demonstrate the robustness and effectiveness of our $\alpha$-Former in challenging scenarios.

## 6 CONCLUSION

In conclusion, we contribute a novel local feature-aware transformer framework called $\alpha$-Former targeting on camouflaged instance segmentation. Observing the camouflaged objects' characteristics, inspired by humans, we introduce traditional descriptors to current camouflaged instance segmentation methods and use traditional descriptor to simulate the process that human find unnatural boundary of camouflaged instance. Moreover, we design a learnable novel binary filter to extract the camouflaged image's local features. To provide the local features to the encoder, we design a feature aggregation adapter to fuse the pre-trained backbone and the local features input. Furthermore, we create an edge-aware feature fusion module to improve the boundary prediction of camouflaged objects, combining multi-level features and employing the ground truth edge as supervision. We also provide the quantitative results and qualitative results of our $\alpha$-Former to show our robustness to different backgrounds. We believe the $\alpha$-Former is a new state-of-the-art for camouflaged instance segmentation, and it can be transferred to applications like medical diagnosis, photorealistic blending, etc.

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

## A    MORE ABLATION STUDIES

### A.1    INFLUENCE OF THE ENCODER LAYERS AND DECODER LAYERS

In the table.6, we compare the influence of using different encoder and decoder layers in our architecture. We can see that with the increase of encoder and decoder layers, the performance will first improve and then maintain a similar performance. So, in our architecture, we use six layers of encoder and three layers of decoder.

Table 6: Comparison with the traditional descriptor, the best results are highlighted in **bold**.

| encoder | decoder | COD10K | | | NC4K | | |
|---|---|---|---|---|---|---|---|
| | | AP | AP50 | AP75 | AP | AP50 | AP75 |
| 1 | 3 | 37.256 | 68.755 | 37.982 | 39.453 | 69.538 | 40.453 |
| 3 | 1 | 38.453 | 70.188 | 39.423 | 40.020 | 70.358 | 41.168 |
| 3 | 3 | 40.421 | 70.861 | 40.453 | 41.093 | 71.592 | 42.048 |
| 3 | 6 | 41.424 | 72.826 | 40.826 | 41.726 | 72.059 | 42.824 |
| 6 | 3 | **42.453** | **72.735** | 41.758 | **42.936** | **72.905** | **43.278** |
| 6 | 6 | 42.187 | 72.682 | 41.744 | 42.921 | 72.723 | 43.168 |
| 6 | 9 | 42.424 | 72.672 | **41.776** | 42.876 | 72.781 | 43.133 |

### A.2    ABLATION STUDIES OF USING DIFFERENT BACKBONE

In the table.7, we compare the performance of using different backbones in our architecture.

Table 7: Comparison with the traditional descriptor, the best results are highlighted in **bold**.

| Backbone | COD10K | | | NC4K | | |
|---|---|---|---|---|---|---|
| | AP | AP50 | AP75 | AP | AP50 | AP75 |
| Resnet-50(Default) | 42.453 | 72.735 | 41.758 | 42.936 | 72.905 | 43.278 |
| Resnet-18 | 36.489 | 67.159 | 37.188 | 37.458 | 68.711 | 38.950 |
| Resnet-101 | 43.188 | 73.725 | 42.713 | 43.794 | 72.313 | 44.484 |
| Vgg-16 | 37.148 | 68.469 | 37.195 | 39.948 | 69.159 | 40.152 |

## B    MORE IMPLEMENT DETAILS

### B.1    MORE DETAILS OF THE FEATURE AGGREGATION ADAPTER

Our feature aggregation adapter uses a tiny initial value to guarantee at the beginning of the training, the output domain is the same as the input image domain. Specifically, we set the mean and the variance value of the convolution weight as 0 and 0.001, and the bias value of the convolution layer as 0. Using the tiny-initialized convolution layer and the skip connection, we can know that the output of the adapter is almost the same as the input at the beginning of the training.

### B.2    MORE DETAILS OF THE EDGE-AWARE FEATURE FUSION MODULE

In this section, we provide more details about our edge-aware feature fusion module. Our edge-aware feature fusion module uses multi-scale features to predict the boundary of the target object. As shown in table.8, we provide the input and output shapes of the different edge prediction blocks.

Table 8: Input and output shape of different edge prediction block

| Block | Input Shape | Output Shape |
|---|---|---|
| $block_5$ | $\frac{H \times W}{32}$ | $\frac{H \times W}{16}$ |
| $block_4$ | $\frac{H \times W}{16}$ | $\frac{H \times W}{8}$ |
| $block_3$ | $\frac{H \times W}{8}$ | $\frac{H \times W}{4}$ |
| $block_2$ | $\frac{H \times W}{4}$ | $\frac{H \times W}{4}$ |

### B.3    MORE DETAILS OF THE PREDICTION HEAD

In this section, we provide more details about our prediction head. We follow the same architecture as OSFormerPei et al. [2022]. As shown in Fig.8. During the training process, we use a fully-connected layer to calculate the location label. At the same time, we use a multi-layer perceptron to calculate the instance-aware parameters. Then we assign positive and negative locations using ground truth. During the testing process, we use a confidence score of the location label to filter ineffective parameters of the instance-aware parameters. Then we use two linear layers to calculate the weight and bias to calculate the segmentation mask. Then we use an up-sampling operation to get the final prediction masks.

## C    MORE VISUALIZATIONS

As shown in Fig.9, we provide more visualizations in this section.

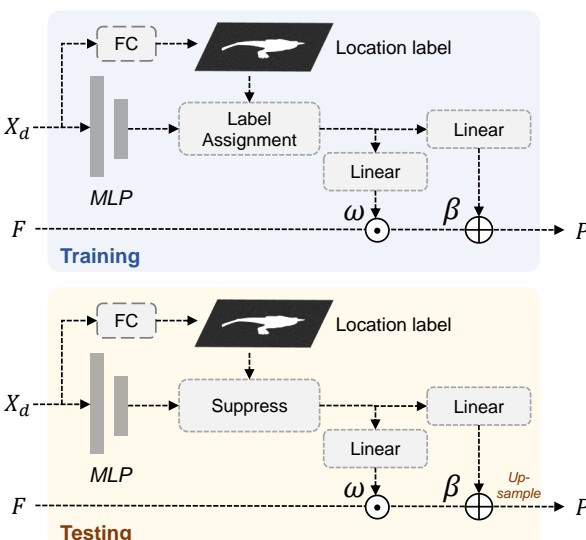

Figure 8: During the training process, our prediction head uses location labels as supervision, and during the testing process, our prediction head uses location labels to filter ineffective parameters.

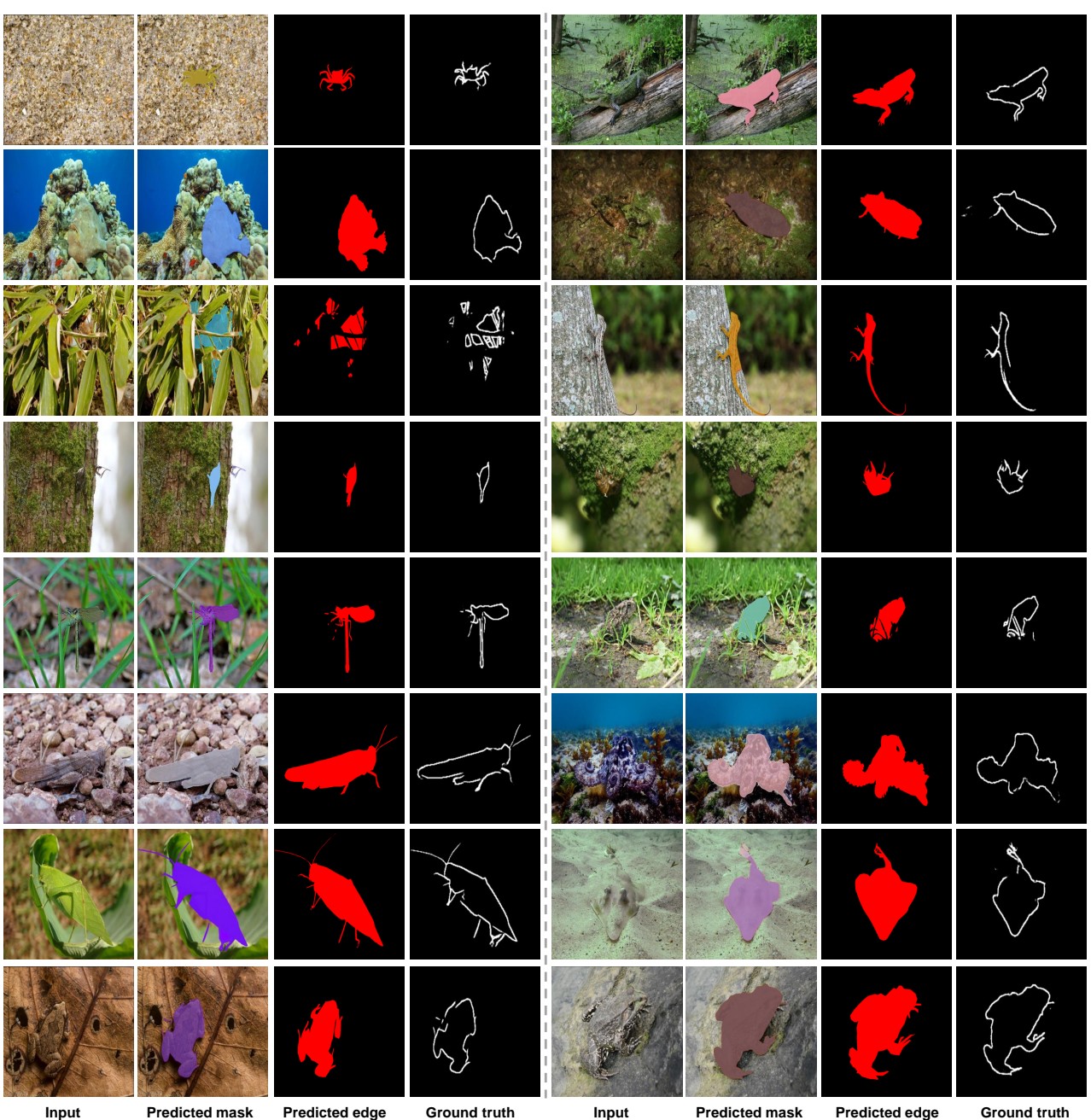

| Input | Predicted mask | Predicted edge | Ground truth | Input | Predicted mask | Predicted edge | Ground truth |

Figure 9: The qualitative results of $\alpha$-Former.