# OpenReview forum: "α-Former: Local-Feature-Aware (L-FA) Transformer"
_auai.org/UAI/2024/Conference — UAI 2024 poster_

### Official Review · Reviewer_Jmwc · 2024-02-26

**Q2-1 Originality-Novelty:** 2
**Q2-2 Correctness-Technical Quality:** 2
**Q2-5 Clarity Of Writing:** 2

**Q1 Summary And Contributions:**

This paper presents $\alpha$-Former, a local-feature-aware transformer for the camouflaged instance segmentation task. The proposed method surpasses state-of-the-art performance on the COD10K and NC4K datasets.

**Q2-3 Extent To Which Claims Are Supported By Evidence:**

2: Fair: the main claims are somewhat supported by evidence (but the experimental evaluation may be weak, or does not match entirely with the claims, important baselines may be missing, proofs contain important ideas but lack rigor, algorithmic details are only discussed superficially, references are imprecise, assumptions are not sufficiently motivated or explicated, etc.).

**Q2-4 Reproducibility:**

2: Fair: key resources (e.g. proofs, code, data) are unavailable but key details (e.g. proof sketches, experimental setup) are sufficiently well-described for an expert to confidently reproduce the main results.

**Q3 Main Strengths:**

(1) The motivation and the solution to address the challenges presented are clearly articulated in the paper.

(2) The qualitative and quantitative results show that the proposed method achieves better empirical performance.

**Q4 Main Weakness:**

(1) The writing quality is overall okay but can be improved. For example, the incorrect citation format makes it hard to distinguish the main content from the citation.

(2) The technical novelty is limited and incremental.

(3) More experiments should be included to support the claims made in the paper, such as using more datasets, comparing with more recent state-of-the-arts, and evaluating with additional metrics.

(4) The method is inspired by how humans find camouflaged instances in given photographs, but there is no analysis or discussion showing that the method works similarly to the human vision system when discovering camouflaged instances.

**Q5 Detailed Comments To The Authors:**

Please see detailed strengths and weaknesses.

**Q9 Complying With Reviewing Instructions:**

Yes

---

> ### Author Rebuttal · Authors · 2024-04-08
>
> Thank you for reviewing our paper. We reply to the comments here。
> 1. Citation Format
>    Thanks for your suggestions. We will revise the citation format in the final version.
> 2. Technical Novelty
>    Our motivation is inspired by the human vision system, which can help the model to better find the camouflaged object. In order to simulate this system, we try to combine the classical filters and transformer to improve the performance of the model. We also add the edge fusion module to help the model better extract the edge of the object. In summary, our model has these technical novelties:
>     1. We observe the human vision system to find the camouflaged object and try to simulate this process.
>     2. We propose a novel camouflaged instance segmentation model that combines classical filters and transformer to improve the performance of the model.
>     3. We use BCNN as an edge fusion module to help the model better extract the edge of the object.
> 3. More experiments
>     1. Here, we try to conduct more experiments to evaluate our model. Here we compare the number of parameters, FLOPs, and inference time of our model with the state-of-the-art methods. All of the methods' backbone is ResNet-50.
>      | Method | Parameters | FLOPs | Inference Time |
>      | --- | --- | --- | --- |
>      | CondInst | 34.1M | 200.1G | 0.052s |
>      | SOTR | 63.1M | 476.7G | 0.109s |
>      | SOLOV2 | 46.2M | 318.7G | 0.081s |
>      | OSFormer | 46.6M | 324.7G | 0.071s |
>      | Ours | 47.2M | 331.9G | 0.072s |
>      2. Here, we provide more evaluation metrics to evaluate our model. Here are some additional metrics to evaluate camouflaged object detection and we using the CHAMELEON dataset to evaluate different models. Here are the results:
>     | Method | $S_\alpha \uparrow$ | $E_\phi \uparrow$  | $E_\beta^w \uparrow$ | $M \downarrow$ |
>     | --- | --- | --- | --- | --- |
>     | EGNet [1] | 0.848 | 0.870 | 0.702 | 0.050 |
>     | PraNet [2] | 0.860 | 0.907 | 0.763 | 0.044 |
>     | SINet-V2 [3] | 0.888 | 0.942 | 0.816 | 0.030 |
>     | Ours | 0.894 | 0.952 | 0.825 | 0.028 |
>
> 4. Human Vision System
>    This is a good point, there are paper[4] that have studied how humans find camouflaged instances in given photographs. This is our original motivation to propose this model. During our process of exploration, we notice that classical fliters work a similar way as how human find camouflaged instances. and experiment results show that the classical filters can help the model better find the camouflaged object. Due to the complexity of human vision system, we cannot directly prove that our model works similarly to the human vision system when discovering camouflaged instances. However, inspired by the human vision system, we try our best to simulate this process and propose a novel model to improve the performance of camouflaged instance segmentation.
>
> References:
> [1] Zhao, Jia-Xing, et al. "EGNet: Edge guidance network for salient object detection." Proceedings of the IEEE/CVF international conference on computer vision. 2019.
> [2] Fan, Deng-Ping, et al. "Pranet: Parallel reverse attention network for polyp segmentation." International conference on medical image computing and computer-assisted intervention. Cham: Springer International Publishing, 2020.
> [3] Fan, Deng-Ping, et al. "Concealed object detection." IEEE transactions on pattern analysis and machine intelligence 44.10 (2021): 6024-6042.
> [4] Troscianko, Tom, et al. "Camouflage and visual perception." Philosophical Transactions of the Royal Society B: Biological Sciences 364.1516 (2009): 449-461.

---

### Official Review · Reviewer_nvHn · 2024-03-10

**Q2-1 Originality-Novelty:** 3
**Q2-2 Correctness-Technical Quality:** 3
**Q2-5 Clarity Of Writing:** 3

**Q1 Summary And Contributions:**

This paper aims at the camouflaged instance segmentation. To address the issue that the target is very similar to the background, this paper proposes a local feature-aware transformer that employs traditional descriptors as prior knowledge. The experiments show that the proposed method achieves better performance than other methods.

**Q2-3 Extent To Which Claims Are Supported By Evidence:**

3: Good: the main claims are supported by convincing evidence (in the form of adequate experimental evaluation, proofs, (pseudo-)code, references, assumptions).

**Q2-4 Reproducibility:**

3: Good: key resources (e.g. proofs, code, data) are available and key details (e.g. proofs, experimental setup) are sufficiently well-described for competent researchers to confidently reproduce the main results.

**Q3 Main Strengths:**

- The proposed method is well-motivated and efficient.

- The proposed method achieves good performance.

- This paper is well-written.

**Q4 Main Weakness:**

- The title is inappropriate. As this paper mainly is focused on camouflaged instance segmentation and it also does not validate its performance on general object segmentation, the authors should refer to the task in the title.

- As shown in Tab. 1, the improvements brought by the proposed method in comparison with previous methods are weak. Therefore, the authors should provide more information for better comparisons, such as the number of parameters, FLOPs, and inference time.

**Q5 Detailed Comments To The Authors:**

See Q4.

**Q9 Complying With Reviewing Instructions:**

Yes

---

> ### Author Rebuttal · Authors · 2024-04-08
>
> Thank you for reviewing our paper. We reply to the comments here。
> 1. Title
>    Thanks for your suggestions. We will try to modify the title to better describe the task of the paper in the final version.
> 2. Performance Comparison
>     Thanks for your suggestions. Here, we try to provide more information for better comparisons. Here are the results of the number of parameters, FLOPs, and inference time of our model. All of the methods' backbone is ResNet-50.
>      | Method | Parameters | FLOPs | Inference Time |
>      | --- | --- | --- | --- |
>      | CondInst | 34.1M | 200.1G | 0.052s |
>      | SOTR | 63.1M | 476.7G | 0.109s |
>      | SOLOV2 | 46.2M | 318.7G | 0.081s |
>      | OSFormer | 46.6M | 324.7G | 0.071s |
>      | Ours | 47.2M | 331.9G | 0.072s |

---

### Official Review · Reviewer_98zP · 2024-03-20

**Q2-1 Originality-Novelty:** 2
**Q2-2 Correctness-Technical Quality:** 3
**Q2-5 Clarity Of Writing:** 3

**Q10 Ethical Concerns:**

No.

**Q1 Summary And Contributions:**

The authors propose a model that uses traditional computer vision descriptors as prior knowledge to improve performance in the camouflaged instance segmentation (CIS) task. They also suggest a learnable module that extracts features similar to these descriptors. Additionally, they introduce edge supervision to estimate sharper boundaries, proving the significance of this method in representative datasets in this field.

**Q2-3 Extent To Which Claims Are Supported By Evidence:**

3: Good: the main claims are supported by convincing evidence (in the form of adequate experimental evaluation, proofs, (pseudo-)code, references, assumptions).

**Q2-4 Reproducibility:**

2: Fair: key resources (e.g. proofs, code, data) are unavailable but key details (e.g. proof sketches, experimental setup) are sufficiently well-described for an expert to confidently reproduce the main results.

**Q3 Main Strengths:**

+ The authors' introduction of their main idea for the CIS problem is very technically valid. Also, the background explanation for this is straightforward to understand.
+ The BCNN module proposed by the authors is very interesting. The idea of introducing a quantized filter to overcome the limitations of transformers is very interesting from a representation learning perspective, as it indirectly exhibits different properties from learning-based features.

**Q4 Main Weakness:**

* The authors' introduction of classical filters looks pretty reasonable. However, I believe there needs to be a clearer explanation of whether this is due to locality preservation or quantization.
  * One of the baselines, OSFormer, also tried to preserve local features through CNN. However, according to Table 2, adding classical filters generally results in better performance. This could be due to the lack of locality, one of the author's motivations, which may have led to the decline in the performance of the existing algorithms. I'd like to ask their opinions on why such features are challenging to acquire through learning.

* Verification of BCNN module
  * Recently, many methods have been proposed for quantizing neural networks. It seems that validation is missing as to whether the method proposed by the author is more suitable than these methods.
  * A comparison experiment is also needed, such as comparing cases where -1, 0, 1 are randomly selected and the binary filter set considering all cases of 3 by 3.

* Authors should apply appropriate citation format to improve the paper's readability.

**Q5 Detailed Comments To The Authors:**

* Can the authors provide insight into why it is difficult for transformers to be trained to extract features directly similar to the LBP descriptor?

* The authors claim that in the CIS task, the CNN structure seems to work more effectively than the transformer structure (see the motivation: human perception system). Among the existing works, the one with the best performance is a model with a wider receptive field, like a transformer. Could you provide an additional explanation on this?

**Q9 Complying With Reviewing Instructions:**

Yes

---

> ### Author Rebuttal · Authors · 2024-04-08
>
> Thank you for reviewing our paper. We reply to the comments here。
> 1. Classical Filters
>    This is a good point of why classical filters can improve the performance of the model. From my understanding, CNN can preserve local features through convolutional operations. However, the learning process of CNN is data-driven, which means that the model can only learn the features that are in the dataset. If the dataset is not large enough, the model may not learn some features that are important for the task. Instead, the model will learn some features that can improve the most performance on the dataset like localizing the object. In this case, the classical filters can provide some prior knowledge to the model, which can provide the model the ability to better find the boundary of the object. This operation can benefit the model and improve the performance. However, this is not the most important features that the model needs to learn. In our experiments, our results show that the classical filters can help the model learn some features that are important for the camouflaged instance segmentation task. In order to better expain this problem, we try to add one more experiments. Here is the experiment setting: during the feature extraction process, the difference between our architecture and OSFormer is that we add the classical filters to the feature extraction process. Then we use the same backbone to extract the features. Here, we try to modify the backbone of OSFormer to add the classical filters to the feature extraction process. We add one BCNN layer to the origianl backbone of OSFormer which will help the OSFormer to extract similar features as classical filters. Then we compare the performance of the two models. Due to the time limit, we only conduct experiment on COD10K dataset. Here are the results:
>     | Method | AP | AP50 | AP75
>     | --- | --- | --- | --- |
>     | OSFormer | 41.0 | 71.1 | 40.8 |
>     | OSFormer with BCNN layer | 41.5 | 71.4 | 41.2 |
>
>     From the results, we can see that the model with BCNN layer can achieve better performance than the original OSFormer. This experiment can prove that the directly use backbone cannot extract all of the features that classical filter can provide.
>
> 2. More Experiments
>     Thanks for your suggestions. Here, we try to conduct such experiments. However, if we want to consider all cases of 3 by 3, the number of filters will be $3^9$, which is too large compared to the original setting. In order to provide better understaind, here, we try to conduct similar experiments, in which we add the kernel numbers of the BCNN layer to see if the model can achieve better performance. In this experiments, we guarantee that every kernel is different. Here are the results:
>     | Kernel number | AP | AP50 | AP75 |
>     | --- | --- | --- | --- |
>     | 64 | 42.5 | 72.8 | 41.8 |
>     | 256 | 42.6 | 72.8 | 41.9 |
>     | 1024 | 42.4 | 72.9 | 41.9 |
>     | 4096 | 42.7 | 73.0 | 42.0 |
>     | 16384 | 42.6 | 72.8 | 41.7 |
>
>     From the results, we can see that 64 kernels are enough for the BCNN layer to achieve the best performance. Adding more kernels's improvement is not significant.
>
> 3. Citation Format
>    Thanks for your suggestions. We will revise the citation format in the final version.

---

### Official Review · Reviewer_N1yr · 2024-03-24

**Q2-1 Originality-Novelty:** 2
**Q2-2 Correctness-Technical Quality:** 3
**Q2-5 Clarity Of Writing:** 3

**Q1 Summary And Contributions:**

This paper introduces a novel approach that merges traditional vision descriptors with modern neural networks to improve edge detection and object localization in camouflaged instance segmentation tasks. The motivation behind this approach is straightforward: leveraging the object boundary can enhance both detection and localization accuracy. The method employs a binary filter, which effectively captures this property and aids in the refinement of segmentation results.

**Q2-3 Extent To Which Claims Are Supported By Evidence:**

3: Good: the main claims are supported by convincing evidence (in the form of adequate experimental evaluation, proofs, (pseudo-)code, references, assumptions).

**Q2-4 Reproducibility:**

2: Fair: key resources (e.g. proofs, code, data) are unavailable but key details (e.g. proof sketches, experimental setup) are sufficiently well-described for an expert to confidently reproduce the main results.

**Q3 Main Strengths:**

- The motivation is clear and technically sound.
- The method is easy to follow.
- The figures are clear to present the model architectures and other necessary information.
- The results are convincing while the ablation studies verify the effect of the proposed modules.

**Q4 Main Weakness:**

- Some model designs are presented without explanation. For example, why adding a CNN architecture in both encoder and decoder after the Transformer. Following the segmentation literature it seems that a transformer decoder is enough to achieve good results.
- Why edge fusion module at the end could further benefit the performance? Since the Binary Filter at start as well as the shallower layers of the backbone should already learn such information and they are already encoded into the representation through the encoder and the decoder.

**Q5 Detailed Comments To The Authors:**

Please see above strengths and weakness.

**Q9 Complying With Reviewing Instructions:**

Yes

---

> ### Author Rebuttal · Authors · 2024-04-08
>
> Thank you for reviewing our paper. We reply to the comments here。
> 1. Model Design
> Yes, only using transformer can already achieve good performance in segmentation tasks. However, compared to CNN, the transformer typically relies on more computing resources and large datasets. However, the camouflaged instance segmentation task doesn't have a very large dataset. Hence, we do not use some classic transformer-based segmentation architecture. Instead, we try to combine CNN and transformer to decrease the request for dataset scale and computing resources.
>
> 1. Edge Fusion Module
> Edge detection poses a significant challenge in current camouflaged instance segmentation methodologies. This challenge stems from the distinctive nature of camouflaged instance segmentation compared to conventional instance segmentation tasks. Identifying the target object is particularly arduous, especially along its boundaries, as it shares similar colors and textures with the background. Although our Binary Filter and backbone can already extract the information of the object edge, without a designed module to optimize the edge extraction, our model will still suffer from extracting some difficult edges. Due to this, we tried to add an edge fusion module and a loss to directly supervise the edge extraction. This additional module significantly enhances the architecture's capability to accurately delineate the edges of the target object.

---

### Meta-Review · Area_Chair_sbAA · 2024-04-16

This paper presents a local feature-aware transformer for the camouflaged instance segmentation task. During the reviewing process, the paper was reviewed by four expert reviewers, and all of them gave positive ratings after the rebuttal period.

The authors are required to update their paper and include the discussions and additional results in the final version. Besides, they need to address `Reviewer Jmwc`'s further concerns, as follows,

> - Address the motivation. As the authors mentioned in the rebuttal, if they cannot prove that their method follows the HVS principle, I would recommend not exaggerating.
> - Add more discussions regarding related work. There is a notable absence of discussion in terms of SOTA methods.
> - Include more experiments to demonstrate the effectiveness of combining a traditional pre-defined feature extractor with CNN and Transformer components (this practice lacks novelty, in my opinion).
> - Provide a comprehensive evaluation. Common datasets for benchmarking camouflaged object detection, such as CAMO and CHAMELEON, should be considered for evaluation.